# OpenReview forum: "Sparse Tokens Suffice: Jailbreaking Audio Language Models via Token-Aware Gradient Optimization"
_ICML.cc/2026/Conference — ICML 2026 regular_

### Official Review · Reviewer_CArn · 2026-02-22

**Soundness:** 3
**Presentation:** 3
**Significance:** 3
**Originality:** 3
**Overall Recommendation:** 4
**Confidence:** 3

**Summary:**

This paper introduces Token-Aware Gradient Optimization (TAGO), a novel and highly efficient white-box jailbreak attack against Audio Language Models (ALMs) that challenges the necessity of traditional dense waveform updates. By analyzing the structure of token-aligned gradients, this work discovered extreme "token-level heterogeneity," meaning that the optimization signal required to manipulate the model is heavily concentrated within a very small subset of audio tokens. Leveraging this insight, TAGO performs sparse optimization by identifying these high-energy tokens and selectively applying adversarial perturbations only to their corresponding waveform regions. To ensure the attack successfully elicits harmful content, the method also forces the model to begin with a model-compatible response prefix and penalizes early termination. Evaluations across three state-of-the-art ALMs, Qwen3-Omni, Qwen2.5-Omni, and LLaMa-Omni, demonstrate that TAGO consistently outperforms prior dense baseline attacks and maintains high attack success rates even when discarding up to 75% of the token updates, effectively proving that full-waveform perturbations are largely redundant for audio jailbreaks.

**Compliance With Llm Reviewing Policy:**

Affirmed.

**Final Justification:**

Final Justifications:
- The authors justify their practical relevance by citing the high download count of open-source models like Qwen3-Omni.
- While developers download these weights, they subsequently deploy them behind APIs or application interfaces. An attacker interacting with a real-world application (like a customer service bot or voice assistant powered by Qwen3-Omni) only has black-box access to the system, rendering a gradient-dependent white-box attack impossible to execute in that environment.
- The authors state that their attack requires gradient access to leverage token-level gradient heterogeneity. For this to be a real-world threat, an attacker would need to generate the adversarial audio on their local, white-box copy of the model, and then successfully transfer that audio to attack a black-box deployed version.
- Since the paper does not demonstrate this black-box transferability, readers can easily argue that the real-world applicability remains unproven.
- Keeping the score as I think this is a good research paper, but very limited in real-world application settings because it is a whitebox threat model.
- This problem is also raised by all reviewers.

**Key Questions For Authors:**

How successful does the proposed sparse-token attack transfer after model updates?

**Limitations:**

yes

**Strengths And Weaknesses:**

Strengths:
- TAGO successfully demonstrates that the optimization space for adversarial audio can be heavily sparsified. By retaining only a small fraction of token updates (e.g., 25%), the method maintains high attack success rates, proving that traditional dense waveform updates are largely redundant.
- This work provided a strong empirical foundation by analyzing the structure of token-aligned gradients. They discovered extreme token-level gradient heterogeneity, revealing that a disproportionately small subset of audio tokens dominates the optimization signal.
- The method is highly effective in practice, outperforming existing dense baseline attacks (such as SpeechGuard and AdvWave) across multiple state-of-the-art Audio Language Models, including Qwen3-Omni, Qwen2.5-Omni, and LLaMa-Omni.

Weaknesses:
- Most commercial models nowadays are released by api calls, hence their method lacks application potential in blackbox scenarios.
- Though this work do take a few basic steps to keep the noise from becoming overwhelmingly obvious, the paper completely lacks actual perceptual evaluations, such as human listening tests (Mean Opinion Score) or standard acoustic quality metrics (like PESQ or STOI), to confirm if a human would notice the adversarial noise when listening to the audio.

---

> ### Author Rebuttal · Authors · 2026-03-26
>
> We sincerely thank the reviewer for the positive assessment and encouraging Weak Accept recommendation. We especially appreciate the recognition of the paper’s main strengths.
>
> # On the white-box setting
> We agree that TAGO is a white-box method and does not directly apply to API-only black-box systems. However, this threat model is deliberate and aligned with the central goal of the paper. Our main question is not only how to obtain a stronger attack, but more importantly how the optimization signal is structured in ALM jailbreaks, and whether dense waveform updates are actually necessary. Answering this question requires gradient access. Accordingly, the paper is fundamentally mechanism-centered: we identify a stable token-level gradient heterogeneity phenomenon in ALMs and show that exploiting this structure enables highly effective sparse optimization. In this sense, the white-box setting is the natural setting for studying the internal optimization mechanism, rather than an assumption introduced merely for convenience. We fully agree that black-box ALM jailbreak is important, but it addresses a different question and is a natural future direction rather than the target of this paper. More specifically, our work first analyzes the white-box setting to reveal and validate token-level gradient heterogeneity in ALM jailbreak optimization. Turning this mechanism into an effective black-box attack is, in our view, an important and valuable future direction.
>
> Besides, the white-box threat model is the standard setting for studying optimization-based jailbreak attacks on language models. Prior work on text-based jailbreaks [1], vision-based jailbreaks [2], and the audio jailbreak baselines we compare against, SpeechGuard and AdvWave, all adopt the same white-box assumption. Our work follows this established setting to ensure a fair and consistent comparison with existing methods.
>
> # On perceptual quality
> We thank the reviewer for pointing out the need for a more explicit perceptual evaluation. To address this concern, we additionally report signal-to-noise ratio (SNR), a standard objective metric in adversarial audio research. Under $\zeta=0.25$ and $\rho=0.9$, the average SNRs are:
>
> |Model|SNR(dB)|
> |---|---:|
> |Qwen3-Omni|20.65|
> |Qwen2.5-Omni|21.83|
> |LLaMA-Omni|22.45|
>
> These results indicate that the perturbations remain relatively imperceptible overall while TAGO still achieves strong jailbreak performance. At the same time, imperceptibility is not the primary goal of this paper. The paper is mainly a mechanism-centered study showing that ALM jailbreak optimization exhibits strong token-level gradient heterogeneity and that exploiting this structure enables highly effective sparse optimization. If stronger imperceptibility is desired, one can further reduce the perturbation budget $\epsilon$ and increase the maximum number of optimization iterations.
>
> # On transfer after model updates
> We interpret this question as asking whether a perturbation optimized on one model version remains effective after the target model is updated. This is closely related to transferability. We therefore additionally report cross-model transfer results under $\rho=0.9$ and $\zeta=0.25$:
>
> |Origin model|Qwen3-Omni|Qwen2.5-Omni|LLaMA-Omni|
> |---|---:|---:|---:|
> |Qwen3-Omni|99/86|31/4|100/46|
> |Qwen2.5-Omni|10/1|97/53|100/37|
> |LLaMA-Omni|4/0|25/1|100/72|
>
> Each cell reports ASR_r/ASR_l. These results show that transferability is poor overall. Therefore, we do not expect previously optimized perturbations to remain strongly effective after nontrivial model updates, especially when the update changes the internal gradient structure. Importantly, this does not contradict the main contribution of the paper. TAGO is not designed as a transfer-oriented attack. Rather, it is a mechanism-centered study showing that ALM jailbreak optimization exhibits strong token-level gradient heterogeneity, and that exploiting this structure enables highly effective sparse optimization in the native white-box setting.
>
> More broadly, weak transferability is unsurprising in the audio adversarial domain, where prior black-box audio attack work also suggests that transfer is substantially more challenging than in many vision settings [3,4]. In the open-source setting considered here, if a new checkpoint is released, TAGO can be directly re-optimized on the updated checkpoint.
>
> # Reference
> [1] Zou, Andy, et al. "Universal and transferable adversarial attacks on aligned language models." arXiv preprint, 2023.
>
> [2] Qi, Xiangyu, et al. "Visual adversarial examples jailbreak aligned large language models." AAAI, 2024.
>
> [3] Zheng, Baolin, et al. "Black-box adversarial attacks on commercial speech platforms with minimal information." CCS, 2021.
>
> [4] Fang, Zheng, et al. "Zero-query adversarial attack on black-box automatic speech recognition systems." CCS, 2024.
>
> We hope the above clarifications resolve your concerns and will be taken into account in the final assessment.

---

> > ### Author Rebuttal · Reviewer_CArn · 2026-04-03
> >
> > - Keeping the score as I think this is a good research paper, but very limited in real-world application settings because it is a whitebox threat model. This problem is raised by all reviewers.

---

> > > ### Author Response · Authors · 2026-04-03
> > >
> > > Thank you for the update and for the positive assessment. We especially appreciate your recognition that this is a good research paper.
> > >
> > > We understand your remaining concern about real-world applicability under the white-box threat model. We would like to briefly reiterate that the main goal of this work is to uncover a mechanism of ALM jailbreak optimization in the white-box setting, namely token-level gradient heterogeneity and the resulting effectiveness of sparse token-aware optimization. Since this question is fundamentally about the internal optimization structure, it requires gradient access and is therefore most naturally studied in the white-box setting.
> > >
> > > At the same time, we believe this setting is still practically relevant, since many strong open-source ALMs are widely studied, deployed, and adapted in real applications. For example, Qwen3-Omni is openly released on Hugging Face and shows substantial community usage, with about 364,939 downloads in the last month at the time of rebuttal. If the paper is accepted, we will make this positioning even more explicit in the final version.
> > >
> > > Thank you again for your careful reading and constructive feedback.

---

### Official Review · Reviewer_WUQa · 2026-03-08

**Soundness:** 3
**Presentation:** 3
**Significance:** 2
**Originality:** 2
**Overall Recommendation:** 4
**Confidence:** 3

**Summary:**

This paper addresses the inefficiency of existing optimization-based jailbreak attacks on audio language models (ALMs). These attacks typically apply dense updates to the entire audio waveform at each iteration. The authors demonstrate that this dense optimization is largely unnecessary, as gradient energy is highly non-uniform and concentrated in a small subset of pre-attention audio tokens.
To address these challenges, the authors propose Token-Aware Gradient Optimization (TAGO), a sparse optimization method that computes token-aligned gradients and selectively updates only the waveform regions corresponding to high-energy tokens, masking the remaining gradients. Furthermore, to maximize attack success in these sparse environments, TAGO constructs model-compatible target prefixes tailored to each ALM's unique response style and incorporates an EOS suppression term to prevent the model from prematurely terminating generation.

**Compliance With Llm Reviewing Policy:**

Affirmed.

**Final Justification:**

This paper makes a meaningful contribution by identifying token-level gradient heterogeneity in ALM jailbreak optimization and proposing a sparse attack method that exploits this structure. The author's response satisfactorily resolves my major concerns regarding model-compatible prefix design and judgment inconsistencies between models.

**Key Questions For Authors:**

- The authors assumed a white-box environment where gradients are computed directly. How can this technique be applied to real-world black-box ALM?

- The paper extracts a specific response style and forces the model to use a fixed prefix. Could this result in contextual inconsistency, which could degrade internal optimization performance or make responses more easily detected by defensive filters?

**Limitations:**

yes

**Strengths And Weaknesses:**

### Strength
- The paper uncovers the root cause of inefficiency in existing audio jailbreaks by revealing that optimization gradients are highly non-uniform and heavily concentrated in a small fraction of pre-attention audio tokens
- The paper improves attack efficiency through smart optimization details, specifically by constructing model-compatible target prefixes tailored to native response styles and incorporating EOS suppression terms to prevent premature termination.
- The paper demonstrates superior attack efficiency, showing that selective updates of sparse tokens (e.g., retaining only 10-25% of tokens) maintain high attack success rates and consistently outperform existing dense baseline attacks on several state-of-the-art ALMs.
### Weakness
- The paper assumes a strict white-box threat model where the attacker has full access to the model's parameters and gradients, which limits its direct application to real-world black-box commercial APIs.
- The paper relies on fixed model-compatible prefix templates that may not perfectly align contextually with every diverse harmful query, which the authors themselves acknowledge as a limitation
- The paper employs different LLM judges (Sorry-Bench and Gemini 3 Flash) for different target ALMs to mitigate false positives, which complicates absolute cross-model vulnerability comparisons.

---

> ### Author Rebuttal · Authors · 2026-03-29
>
> We sincerely thank the reviewer for the careful reading and constructive feedback. We especially appreciate the recognition of the paper’s main strengths.
>
> # On the white-box setting
> We clarify that TAGO is a white-box method and does not directly apply to API-only black-box systems. However, this threat model is deliberate and aligned with the central goal of the paper. Our main question is not only how to obtain a stronger attack, but more importantly how the optimization signal is structured in ALM jailbreaks, and whether dense waveform updates are actually necessary. Answering this question requires gradient access. Accordingly, the paper is fundamentally mechanism-centered: we identify a stable token-level gradient heterogeneity phenomenon in ALMs and show that exploiting this structure enables highly effective sparse optimization. In this sense, the white-box setting is the natural setting for studying the internal optimization mechanism, rather than an assumption introduced merely for convenience. We fully agree that black-box ALM jailbreak is important, but it addresses a different question and is a natural future direction rather than the target of this paper. More specifically, our work first analyzes the white-box setting to reveal and validate token-level gradient heterogeneity in ALM jailbreak optimization. Turning this mechanism into an effective black-box attack is, in our view, an important and valuable future direction.
>
> Besides, the white-box threat model is the standard setting for studying optimization-based jailbreak attacks on language models. Prior work on text-based jailbreaks (Zou et al., 2023), vision-based jailbreaks (Qi et al., 2024), and the audio jailbreak baselines we compare against, SpeechGuard (Peri et al., 2024) and AdvWave (Kang et al., 2024), all adopt the same white-box assumption. Our work follows this established setting to ensure a fair and consistent comparison with existing methods.
>
> # On the model-compatible prefix
> We would like to clarify that the prefix construction is not entirely manual or ad hoc. As described in Section 5.3, TAGO derives the prefix template by querying the target ALM with a small set of benign prompts and extracting the first sentence of the model’s own completions. This procedure is systematic, repeatable, and requires only a few forward passes per model. The key idea is to align the optimization target with the model’s native response style rather than imposing an externally crafted prefix (e.g., “Sure, here is...”), which may be stylistically misaligned with the model’s typical output distribution.
>
> This also addresses the concern about contextual inconsistency. If the fixed prefix were strongly mismatched to the model’s preferred response style, we would expect degraded optimization performance. Empirically, however, the model-compatible prefix consistently improves attack effectiveness, suggesting that it better matches the model’s native generation style rather than introducing an unnatural template.
>
> Furthermore, we note that the ablation study in Table 9 and Figure 5 directly evaluates this design choice. Under ρ=0.9 and ζ=0.25, the model-compatible prefix consistently and substantially outperforms a fixed, model-agnostic prefix across all three ALMs. For example, on Qwen2.5-Omni, ASR_l improves from 26% to 53%, and on LLaMA-Omni, from 48% to 72%. These results indicate that even a simple model-compatible template provides meaningful gains over the common practice of using a hand-crafted fixed prefix, and that the template does generalize well across diverse harmful queries within each ALM.
>
> We acknowledge that a single template may not be optimal for every possible query context. As noted in our Limitation section (Section 7), we value replacing the prefix-constrained objective with adaptive objectives as a promising direction for future work.
>
> # On the use of different judges
> We would like to clarify that we used Gemini 3 Flash for LLaMA-Omni because we found that the Sorry-Bench judge performs poorly on LLaMA-Omni outputs in that setting. For each specific ALM, however, the judge is kept the same across all attack methods, so the comparisons between TAGO and the baselines on that ALM remain fair.
>
> We also clarify that different judges mainly complicate absolute cross-model comparisons, rather than the within-model comparison central to our claim. To further address this concern, we additionally performed a unified evaluation using Qwen3Guard for TAGO under ρ=0.9 and ζ=0.25:
>
> |Model|TAGO(Qwen3Guard)|
> |---|---:|
> |Qwen3-Omni|87|
> |Qwen2.5-Omni|70|
> |LLaMA-Omni|97|
>
> These unified-judge results remain consistent with the overall conclusion that LLaMA-Omni is the least aligned among the three models, while TAGO remains highly effective across ALMs. We will clarify this point in the rebuttal and final version.
>
> We hope the above clarifications resolve your concerns and will be taken into account in the final assessment.

---

> > ### Author Rebuttal · Reviewer_WUQa · 2026-04-04
> >
> > Thank you for the detailed author response. The rebuttal has addressed several of my main concerns. In particular, the clarification on the model-compatible prefix construction and the supporting ablation results (Table 9) are convincing, and the unified judge evaluation using Qwen3Guard helps mitigate my concern about cross-model comparison fairness.
> >
> > One remaining suggestion for the final version: the paper may benefit from discussing Liu et al. (AAAI 2020), "Weighted-Sampling Audio Adversarial Example Attack," which shares a conceptually related idea of selective perturbation in the audio domain. While TAGO's dynamic, gradient-energy-based token selection is clearly distinct from their approach, acknowledging this prior work would strengthen the related work section.
> > Overall, I am updating my score.

---

> > > ### Author Response · Authors · 2026-04-04
> > >
> > > Thank you very much for the update and for the positive acknowledgement. We greatly appreciate your careful reading of our rebuttal and your encouraging comments.
> > >
> > > We are especially grateful that the clarification on the model-compatible prefix construction and the supporting ablation results helped address your concern, and that the unified Qwen3Guard evaluation also alleviated your concern about cross-model comparison fairness.
> > >
> > > We also sincerely thank you for the helpful suggestion regarding Liu et al. (AAAI 2020), *Weighted-Sampling Audio Adversarial Example Attack*. We agree that this is relevant prior work to acknowledge. While TAGO differs in using dynamic token-level selection based on gradient energy during jailbreak optimization, we will discuss this paper in the related work section in the final version if the paper is accepted.
> > >
> > > Thank you again for your constructive feedback and for updating your score.

---

### Official Review · Reviewer_XnKS · 2026-03-10

**Soundness:** 3
**Presentation:** 3
**Significance:** 3
**Originality:** 3
**Overall Recommendation:** 4
**Confidence:** 4

**Summary:**

The paper proposed Jailbreak attacks for audio language models (ALMs) to generate unsafe generations.
The proposed paper finds that gradient energy is non-uniform across audio tokens which leads to the fact that a small subset of audio tokens can dominate the optimization signal.
The paper proposed Token-Aware Gradient Optimization (TAGO), that uses sparse jailbreak optimization on the audio tokens that have high gradient energy. When comparing the proposed model with other baseline the TAGO algorithm gets better results.

**Compliance With Llm Reviewing Policy:**

Affirmed.

**Final Justification:**

The rebuttal addressed my main concerns, my evaluation keep unchanged - Weak accept

**Key Questions For Authors:**

Questions:
1. It will be very interesting to see the results on getting the sparse perturbations optimized on one model and testing it on another model. Does the algorithm still work?
2. Can you develop a defence algorithm to monitor these high energy tokens to remove attacks like this?
3. Is there a connection between the downsampling rate of audio encoder and the gradient concentration degree?

**Limitations:**

--

**Strengths And Weaknesses:**

Strength:
1. The proposed model gets state of the art results on multiple architectures such as Qwen3-Omni, Qwen2.5-Omni, and LLaMA-Omni on two different datasets, AdvBench-50 and HarmBench.
2. The paper gives a novel insight about the internal optimization of ALM. which identifies that the gradient energy is concentrated in small amounts of audio tokens.
3. The paper gives both theoretical justification and empirical observation regarding token-level gradient sparsity.

Weakness:
1. The paper gives a model compatible prefix template to improve the results but it is manual and heuristic and may not be optimal for every possible harmful query context.
2. The proposed method has limitations which requires a white box setting which is not applicable against many commercial audio platforms.
3. The paper needs to evaluate the perceptual quality of the output.

---

> ### Author Rebuttal · Authors · 2026-03-29
>
> We sincerely thank the reviewer for the positive assessment and the encouraging Weak Accept recommendation. We especially appreciate the recognition of the paper’s main strengths.
>
> # On the model-compatible prefix
>
> We thank the reviewer for raising this important point. We would like to clarify that the prefix construction is not entirely manual or ad hoc. As described in Section 5.3, TAGO derives the prefix template by querying the target ALM with a small set of benign prompts and extracting the first sentence of the model's own completions. This procedure is systematic, repeatable, and requires only a few forward passes per model. The key idea is to align the optimization target with the model's native response style rather than imposing an externally crafted prefix (e.g., "Sure, here is…"), which may be stylistically misaligned with the model's typical output distribution.
>
> Furthermore, we note that the ablation study in Table 9 and Figure 5 directly evaluates this design choice. Under ρ=0.9 and ζ=0.25, the model-compatible prefix consistently and substantially outperforms a fixed, model-agnostic prefix across all three ALMs. For example, on Qwen2.5-Omni, ASR_l improves from 26% to 53%, and on LLaMA-Omni, from 48% to 72%. These results indicate that even a simple model-compatible template provides meaningful gains over the common practice of using a hand-crafted fixed prefix, and that the template does generalize well across diverse harmful queries within each ALM.
>
> We acknowledge that a single template may not be optimal for every possible query context. As noted in our Limitation section (Section 7), we value replacing the prefix-constrained objective with adaptive objectives as a promising direction for future work.
>
> # On the white-box setting, perceptual quality, and transferability
> We thank the reviewer for these important questions. Since these concerns are also shared by Reviewer CArn, we provide the detailed discussion there. Briefly, the paper is intentionally a mechanism-centered white-box study; we additionally report SNR-based perceptual quality results and cross-model transfer results, which show reasonably imperceptible perturbations but limited transferability across models.
>
> # On possible defenses
> We appreciate the reviewer’s defense-oriented question. The mechanism uncovered in this paper can also inspire defenses. We therefore conducted a preliminary analysis under ρ=0.9 and ζ=0.25. In particular, **High-grad token defense** masks perturbations on the token-aligned regions with the largest accumulated gradient mass.
>
> | Model | Original | Downsampling (16k→12k→16k) | Local smoothing | High-grad token defense |
> |---|---:|---:|---:|---:|
> | Qwen3-Omni | 99 / 86 | 43 / 25 | 45 / 21 | 4 / 1 |
> | Qwen2.5-Omni | 97 / 53 | 73 / 35 | 71 / 34 | 31 / 5 |
> | LLaMA-Omni | 100 / 72 | 100 / 49 | 100 / 50 | 98 / 45 |
>
> Each cell reports `ASR_r / ASR_l`. These results suggest that simple preprocessing defenses such as downsampling and local smoothing can partially reduce attack success, especially on the Qwen models, while directly suppressing the highest-gradient regions yields much larger reductions on Qwen3-Omni and Qwen2.5-Omni. This supports our central claim that these concentrated high-energy regions are causally important to the attack. We emphasize that the last defense is an **oracle-style** analysis rather than a directly deployable defense, since a defender usually cannot access the attack-time accumulated gradients. However, it provides an initial validation that practical defenses may be built around proxy signals for salient regions.
>
> # On the connection to encoder downsampling rate
> This is a very insightful question. We currently do not claim a definitive causal relationship, but our observations suggest that downsampling may have some effect while being unlikely to be the only factor.
>
> In our models, the encoder downsampling rate roughly follows
> Qwen3-Omni > Qwen2.5-Omni = LLaMA-Omni.
> By contrast, the gradient concentration statistics do not follow a single monotonic trend: according to CV, the concentration degree is
> Qwen3-Omni < Qwen2.5-Omni < LLaMA-Omni,
> while according to top-1 token mass, it is
> Qwen3-Omni > LLaMA-Omni > Qwen2.5-Omni.
>
> Therefore, we believe downsampling may influence token granularity and thus partially affect gradient concentration, but the phenomenon is likely also shaped by model-specific training dynamics and representation behavior. In this sense, it may be more similar to other model-internal phenomena such as attention sinks, rather than being determined by downsampling rate alone. We agree that this is an important direction and will highlight it as future work.
>
>
> We hope the above clarifications resolve your concerns and will be taken into account in the final assessment.

---

> > ### Author Rebuttal · Reviewer_XnKS · 2026-04-02
> >
> > Thank you for addressing my concerns

---

> > > ### Author Response · Authors · 2026-04-03
> > >
> > > Thank you for the update and for reading our rebuttal carefully. We appreciate your note that our response addressed your concerns. We also note that you selected “(c) Partially resolved or unresolved, but the remaining concerns are not easily addressed in a short rebuttal,” but did not specify which concrete concerns you still view as unresolved.
> > >
> > > We would therefore like to respectfully clarify that we believe the concrete concerns raised in the review were addressed in the rebuttal, either through clarification of the paper’s intended scope or through additional evidence and experiments. In particular:
> > >
> > > - **Model-compatible prefix:** We clarified that the prefix is systematically derived from benign completions of the target ALM rather than manually crafted for each harmful query, and we pointed to the ablation results showing that it consistently outperforms a naive fixed prefix.
> > >
> > > - **White-box setting:** We clarified that the paper is intentionally a mechanism-centered white-box study, and that understanding the optimization structure of ALM jailbreaks and the necessity of dense waveform updates requires gradient access.
> > >
> > > - **Perceptual quality:** We added SNR-based perceptual evaluation to provide explicit evidence that the perturbations remain reasonably imperceptible under the evaluated setting.
> > >
> > > - **Cross-model transferability:** We added cross-model transfer experiments to directly answer the reviewer’s question. At the same time, we clarified that transferability itself is not the target of this paper: transferable audio jailbreak is outside the main scope of this work, whose focus is the mechanism of sparse optimization in the native white-box setting.
> > >
> > > - **Possible defenses:** We added defense-oriented analyses, including preprocessing-based defenses and a high-gradient-token suppression analysis, to examine how the identified high-energy regions may support future defense design.
> > >
> > > - **Connection to encoder downsampling:** We added further discussion and evidence on the possible connection between encoder downsampling and gradient concentration, clarifying that downsampling may have some effect, but is unlikely to be the sole explanation.
> > >
> > > We understand that some broader questions may still merit fuller discussion in the paper itself, and we will make these points more explicit in the final version if accepted.

---

### Official Review · Reviewer_EW6e · 2026-03-11

**Soundness:** 2
**Presentation:** 3
**Significance:** 2
**Originality:** 3
**Overall Recommendation:** 3
**Confidence:** 4

**Summary:**

This paper investigates the gradient structure underlying white-box jailbreak optimization in audio language models (ALMs). The authors discover that token-aligned gradient energy during iterative perturbation optimization is highly non-uniform — on Qwen3-Omni, the top 16% of audio tokens capture 90% of gradient energy. Based on this observation, they propose TAGO (Token-Aware Gradient Optimization), which at each iteration masks out gradients from low-energy token regions and updates only the waveform samples aligned with high-energy tokens. TAGO additionally constructs model-compatible target prefixes from benign completions and includes an EOS suppression term to prevent premature generation termination. Evaluated on three ALMs (Qwen3-Omni-30B, Qwen2.5-Omni-7B, LLaMA-Omni-8B) with AdvBench-50 and HarmBench, TAGO with 25% token retention achieves ASR_l of 86% on Qwen3-Omni compared to 87% with dense updates. Post-hoc sparsification is shown to be ineffective (1% ASR_l), confirming that sparsity must be enforced during optimization.

**Compliance With Llm Reviewing Policy:**

Affirmed.

**Key Questions For Authors:**

1. Although it is an attack paper, what defense methods can be applied to this specific method? Have the authors considered any existing defenses / or unique defense that can be implemented? How does TAGO perform against audio defense mechanisms? Specifically, does the sparse perturbation structure make TAGO perturbations easier or harder to detect than dense perturbations?

2. The gradient heterogeneity finding is interesting, but the paper treats it as a black-box observation without investigating its cause. Is the sparsity driven by the acoustic feature extractor (Whisper), the temporal downsampling ratio, or the content of the audio (e.g., are high-gradient tokens aligned with speech onset, keyword boundaries, or silence)?

3. Why is the gap between ASR_r and ASR_l so large in some configurations (e.g., SpeechGuard on Qwen3-Omni: 100% ASR_r but only 42% ASR_l)? Does the model produce non-refusal responses that are still not meaningfully harmful, or is the LLM judge overly conservative?

**Limitations:**

1. Although mentioned in the Related Works, the paper also omits any discussion of black-box audio jailbreaks, only comparing with two white-box baselines. While TAGO is a white-box method, positioning it relative to the growing black-box literature is necessary to justify the threat model's relevance. If black-box attacks already achieve high ASR without gradient access, the practical value of a more efficient white-box attack diminishes. The paper should explicitly discuss when and why white-box gradient-based attacks matter given the existence of effective black-box alternatives.

2. AdvBench-50 contains only 50 unique harmful queries (100 audio samples with two TTS speakers). The HarmBench evaluation uses 200 samples. For a paper whose central claim is about gradient structure, not a new attack per se, this limited scale is a weakness. Questions about generalization across different audio content types (conversational, accented, noisy, multi-speaker), different TTS engines, and different query distributions cannot be answered with 50-200 queries. Have the authors considered doing a larger scale prompts?

3. Two of three models are from the Qwen family (Qwen3-Omni, Qwen2.5-Omni). LLaMA-Omni is the only non-Qwen model, and it's notably the weakest. Direct attack already achieves 49% ASR_l on LLaMA-Omni versus 0-1% on the Qwen models, suggesting weak safety alignment. The absence of any proprietary model (GPT-4o-audio, Gemini) is understandable given the white-box constraint, but other open-source ALMs like MiniCPM-o, or Kimi-Audio would diversify the evaluation. The claim that "dense updates are largely redundant" needs validation beyond one model family and one weakly-aligned model.

**Strengths And Weaknesses:**

**1. The gradient heterogeneity finding is well-characterized and convincing.** The token-level gradient energy analysis in Section 4 is thorough. The metrics (CV, top-q mass, q_α) are well-defined and reported across multiple ALMs in Tables 1 and 8. The finding that gradient energy concentrates heavily on a small token subset is not just an aggregate statistic — Figure 2 and 7 show it persists across iterations, with high-energy tokens remaining dominant over time.

**2. The post-hoc pruning result is the most insightful contribution.** Showing that post-hoc sparsification (optimize densely, then prune) yields only 9% ASR_r / 1% ASR_l on Qwen3-Omni, while TAGO with the same sparsity (ζ=0.25) achieves 99% / 86%, is a clean and surprising result. It demonstrates that sparsity fundamentally reshapes the optimization trajectory, not just the final perturbation. This has real implications for understanding adversarial optimization in the audio domain.

**3. The model-compatible prefix construction (Section 5.3) is a practical contribution.** Querying the ALM with benign prompts and extracting the first sentence as a reusable template is simple but effective. The ablation in Table 9 shows substantial gains over fixed prefixes (e.g., 86% vs. 71% ASR_l on Qwen3-Omni). This technique is model-agnostic and could benefit other prefix-constrained attack methods beyond TAGO.

---

> ### Author Rebuttal · Authors · 2026-03-31
>
> We sincerely thank the reviewer for the detailed and thoughtful feedback. We would like to clarify the following points and will make them explicit in the final version if the paper is accepted.
>
> # On defenses and detectability
> Since this concern is also raised by Reviewer **XnKS**, we provide the detailed defense results there and only summarize the key point here. Under simple preprocessing defenses, sparse perturbations drop slightly more than dense ones. This is unsurprising: these are generic acoustic transformations rather than explicit detectors, and sparse perturbations concentrate their effective changes on fewer waveform regions, so signal processing removes a slightly larger fraction of their effective perturbation mass.
>
> # On the cause of gradient heterogeneity
> This is a very insightful question. A deeper causal analysis is a valuable future direction. Our current evidence suggests that the sparsity pattern is unlikely to be explained by a single factor alone. Different ALMs exhibit concentration on different regions, while gradients near the beginning and end of speech content also tend to be relatively large. We believe that the phenomenon is shaped primarily by model training and learned representation behavior, rather than by downsampling alone.
>
> # On the gap between ASR_r and ASR_l
> We manually checked such cases and found that the gap mainly arises because ASR_r is a weak non-refusal metric, while ASR_l requires the response to be meaningfully harmful. In many cases, the output does not begin with an explicit refusal and is therefore counted by the reject-word heuristic, but it still does not provide actionable harmful content. We will clarify in the final version.
>
> # On the relevance of the white-box threat model
> Since the same point is also raised by Reviewer **CArn**, we only summarize the key point here. Our paper is primarily a **mechanism-centered white-box study**: the goal is to reveal token-level gradient heterogeneity and test whether dense waveform updates are actually necessary, which requires gradient access. We clarify this positioning here and will make it explicit in the camera-ready version if accepted. Besides, black-box ALM jailbreaks are certainly important, but current methods often rely on very large numbers of sampled augmentations or queries.
>
> # On evaluation scale and data diversity
> Broader evaluation would certainly strengthen the paper. At the same time, we would like to clarify the current scale: across AdvBench-50 and HarmBench, we evaluate 250 harmful instructions in total, and AdvBench-50 is synthesized with two different TTS speakers.
>
> To further probe robustness beyond the original clean TTS inputs, we conduct 3 supplementary evaluations under ρ=0.9 and ζ=0.25: (1) adding uniform noise U[-0.01,0.01] to the original 100 AdvBench audio samples; (2) synthesizing the 50 AdvBench instructions with 6 Gemini-TTS speakers covering different gender, accent, and emotion settings, yielding 300 samples; and (3) synthesizing the same 50 instructions with 2 CosyVoice speakers (one male, one female), yielding 100 samples. Due to time constraints, we report the full evaluation on Qwen2.5-Omni and LLaMA-Omni, and additionally verify the Gemini multi-speaker setting on Qwen3-Omni.
>
> |Model|Original|Noise|Multi-speaker|CosyVoice|
> |---|---:|---:|---:|---:|
> |Qwen2.5-Omni|97/53|96/56|94.33/55.67(std 3.79/2.08)|99/65|
> |LLaMA-Omni|100/72|100/65|100/71(std 0/3.03)|100/61|
>
> For Qwen3-Omni, the Gemini multi-speaker result is 99/85(std 1.1/5.76).
>
> Each cell reports ASR_r/ASR_l. These results show that TAGO remains robust under noise, speaker variation, and a different TTS engine.
>
> # On model diversity and additional ALMs
> ALMs typically consist of an audio encoder and a language model, and some suggested systems are architecturally less diverse than they may initially appear:
>
> |Model|Audio Encoder|Language model|
> |---|---|---|
> |Qwen3-Omni|AuT|Qwen3-30B-A3B|
> |Qwen2.5-Omni|Whisper|Qwen2.5-7B|
> |LLaMA-Omni|Whisper|Llama-3.1-8B|
> |MiniCPM-o 4.5|Whisper|Qwen3-8B|
> |Kimi-Audio|Whisper|Qwen2.5-7B|
>
> Thus, MiniCPM-o and Kimi-Audio do not introduce a completely different frontend-backend pattern from the models already studied. To further address this point, we added an experiment on MiniCPM-o 4.5:
>
> |Direct|TAGO(ζ=1.0)|TAGO(ζ=0.25)|
> |---|---:|---:|
> |31/0|98/89|74/60|
>
> Each cell reports `ASR_r / ASR_l`. TAGO remains effective on this additional open-source ALM, and sparse optimization ($\zeta=0.25$) still preserves a substantial fraction of the dense attack success.
>
> We also computed gradient heterogeneity statistics on MiniCPM-o 4.5 (average 45.48 audio tokens). For summed gradients: CV = 3.04, TM1 = 36.08%, TM3 = 71.53%, TM5 = 84.65%, TM10 = 94.85%, q0.8 = 4.55, q0.9 = 6.86. This further supports that strong token-level gradient concentration is present beyond the three original models.
>
> We hope the above clarifications resolve your concerns and will be taken into account in the final assessment.

---

> > ### Author Rebuttal · Reviewer_EW6e · 2026-04-03
> >
> > Thank you for the review. Some of my main concerns have been resolved such as the gap between different ASR metric and the MiniCPM-o addition is a good-faith effort.
> >
> > On the other hand, I am still curious about the gradient heterogeneity cause as it would have been a stronger paper if a simple mechanistic analysis was provided beyond the hypothesis as the authors responded.

---

> > > ### Author Response · Authors · 2026-04-05
> > >
> > > We thank the reviewer for the follow-up question. In our initial rebuttal, we already suggested that the observed sparsity is unlikely to be explained by a single factor alone. To address this point more concretely, we conducted two additional analyses.
> > >
> > > ## Component attribution: the effect is not explained by the frontend or downsampling alone
> > >
> > > We compared three ALMs with different combinations of frontend, language backbone, and downsampling ratio:
> > >
> > > |Model|Audio Encoder|Language Model|Downsampling ratio|CV|TM1|TM3|q0.9|
> > > |---|---|---|---:|---:|---:|---:|---:|
> > > |Qwen3-Omni|AuT|Qwen3-30B-A3B|8|2.74|23.43%|56.84%|9.64|
> > > |Qwen2.5-Omni|Whisper|Qwen2.5-7B|2|3.75|12.86%|35.43%|28.05|
> > > |LLaMA-Omni|Whisper|Llama-3.1-8B|2|4.40|15.90%|43.09%|23.81|
> > >
> > > These results support three points:
> > > - Strong gradient concentration persists across all three ALMs.
> > > - Even models sharing the same Whisper frontend and the same downsampling ratio still differ substantially. For example, Qwen2.5-Omni and LLaMA-Omni both use Whisper with ratio 2, yet `TM1` and `q0.9` still differ substantially.
> > > - Qwen3-Omni also exhibits clear heterogeneity despite using a different frontend and a much higher downsampling ratio.
> > >
> > > At the same time, raw token-level comparisons between Qwen3-Omni and the other two models should be interpreted with care, because Qwen3-Omni uses a 4x coarser temporal tokenization (`8` vs. `2`). Under a rough normalization, its `q0.9=9.64` corresponds to about `38.56` token-equivalents, while its `TM1=23.43%` and `TM3=56.84%` become about `5.86%` and `14.21%`. Under this crude adjustment, Qwen3-Omni is not more concentrated than Qwen2.5-Omni or LLaMA-Omni, and in fact appears less concentrated. Therefore, the key conclusion is not that Qwen3-Omni is strongest, but that strong heterogeneity persists across all three ALMs and cannot be explained by frontend or downsampling alone.
> > >
> > > Taken together, the phenomenon is neither Whisper-specific nor reducible to temporal compression alone. A more plausible interpretation is that gradient heterogeneity emerges from the interaction between acoustic tokenization and downstream learned representations.
> > >
> > > ## Content and positional analysis: high-gradient tokens are often aligned with silent regions, but the positional pattern is model-dependent
> > >
> > > We further partitioned tokens into beginning/end/middle regions and into voiced vs. silent tokens. A token is labeled as silent if its energy is more than 40 dB below the per-utterance maximum.
> > >
> > > The results show a strong and highly consistent trend. Under `ρ=0.9, ζ=0.25`, voiced tokens have lower average gradient than silent tokens in `98/100` records for Qwen3-Omni, `99/100` for Qwen2.5-Omni, and `99/100` for LLaMA-Omni. The global mean gradient ratio with silent normalized to `1` is only `0.0233`, `0.0224`, and `0.0001`, respectively. The same pattern remains under dense optimization (`ζ=1.0`), with voiced/silent ratios `0.0074`, `0.0044`, and approximately `0.0000`. Thus, gradient mass is often amplified in low-energy or silent regions rather than only on obviously salient speech segments.
> > >
> > > This is not explained by silent tokens being more numerous. Under `ρ=0.9, ζ=0.25`, the average numbers of voiced/silent tokens per record are `53.44/6.28` for Qwen3-Omni, `202.34/26.98` for Qwen2.5-Omni, and `202.36/26.96` for LLaMA-Omni. Silent tokens are the minority class in all three models, yet they still dominate in average gradient mass.
> > >
> > > The positional pattern is also clearly model-dependent. Under `ρ=0.9, ζ=0.25`, the global mean gradient ratio across beginning/end/middle regions (normalized by `middle=1`) is:
> > > - Qwen3-Omni: `start=4.4394`, `end=28.7763`, `middle=1.0000`
> > > - Qwen2.5-Omni: `start=0.0164`, `end=14.3223`, `middle=1.0000`
> > > - LLaMA-Omni: `start=0.0019`, `end=0.4617`, `middle=1.0000`
> > >
> > > Thus, Qwen3-Omni and Qwen2.5-Omni place particularly large gradient mass near the tail, whereas LLaMA-Omni concentrates more in the middle. If the effect were mainly caused by generic speech landmarks or a fixed tokenizer artifact, we would expect a more uniform positional pattern across ALMs. Instead, even two models sharing the same Whisper frontend and the same downsampling ratio prioritize different temporal regions, suggesting that this preference is more likely shaped by learned model behavior acquired during training.
> > >
> > > Overall, while this does not yet provide a complete theory, it gives a more concrete answer to the reviewer’s question: **the observed sparsity is not attributable to a single frontend or downsampling factor, and its temporal/content alignment is model-dependent, with many high-gradient tokens occurring in low-energy or silent regions rather than only at obvious speech boundaries.** We will incorporate this clarification into the final version and position a deeper mechanistic study as an important direction for future work.
> > >
> > > **We hope the above clarifications resolve your concerns and will be taken into account in the final assessment.**

---

### Decision · Program_Chairs · 2026-04-30

**Decision:**

Accept (regular)

**Comment:**

After the rebuttal, all Reviewers are recommending acceptance.

The rebuttal fully resolved Reviewer WUQa’s concerns. Reviewer XnKS confirmed the Authors "answered [their] questions, and [...] would like to keep my grade as it is". Reviewer CArn maintained their weak accept, acknowledging "this is a good research paper" while noting the white-box limitation (a constraint shared with related works). Reviewer EW6e did not update their rebuttal, but stated in the discussion phase that "Since the reviewers addressed my comments after the feedback, I would agree with leaning towards acceptance".

Given the unanimously positive feedback, I am fully endorsing for acceptance. I would like to encourage the Authors to incorporate the received feedback in the final revision.